# Design of an Integrated System for Spaceborne SAR Imaging and Data Transmission

**DOI:** 10.3390/s24196375

**Published:** 2024-10-01

**Authors:** Qixing Wang, Peng Gao, Zhuochen Xie, Jinpei Yu

**Affiliations:** 1School of Information Science and Technology, ShanghaiTech University, Shanghai 200120, China; wangqx@shanghaitech.edu.cn; 2Innovation Academy for Microsatellites of CAS, Shanghai 201210, China; xiezc.ac@hotmail.com; 3State Grid Electric Power Research Institute, NariARI Group Co., Ltd., Nanjing 211106, China; gp19970106@126.com

**Keywords:** synthetic aperture radar, data transmission, integrated sensing and communication, filter bank multicarrier, optimal design

## Abstract

In response to the conflicting demands between real-time satellite communication and high-resolution synthetic aperture radar (SAR) imaging, we propose a method that aligns the data transmission rate with the imaging data volume. This approach balances SAR performance with the requirements for real-time data transmission. To meet the need for mobile user terminals to access real-time SAR imagery data of their surroundings without depending on large traditional ground data transmission stations, we developed an application system based on filter bank multicarrier offset quadrature amplitude modulation (FBMC-OQAM). To address the interference problem with SAR signals’ transmission and reception, we developed a signal sequence based on spaceborne SAR echo and data transmission and reception. This system enables SAR and data transmission signals to share the same frequency band, radio frequency transmission system, and antenna, creating an integrated sensing and communication system. Simulation experiments showed that, compared to the equal power allocation scheme for subcarriers, the echo image signal-to-noise ratio (SNR) improved by 2.79 dB and the data transmission rate increased by 24.075 Mbps.

## 1. Introduction

Spaceborne synthetic aperture radar (SAR) provides high-resolution Earth observation capabilities in all weather conditions and at all times. It has significant potential for applications such as battlefield surveillance, environmental protection, disaster monitoring, ocean observation, and geological mapping [1,2,3]. As communication networks evolve to space–air–ground integrated systems and deploy low Earth orbit satellite constellations, electromagnetic spectrum resources in space become scarce. SAR satellites have licensed spectrum resources from L-band to Ku-band. Utilizing these quality resources for radar-communication spectrum sharing is a key focus in satellite communication research [4]. Furthermore, satellite SAR remote sensing heavily depends on ground-based relay transmission facilities. In remote and offshore areas with limited infrastructure, mobile terminals are needed to receive real-time imagery data of surrounding regions. However, the lack of traditional large ground-based data transmission stations can cause communication congestion and delays in transmitting large numbers of imagery data. Additional communication links between ground data receiving stations and mobile users are essential for data distribution, but they can introduce delays in obtaining images of time-sensitive targets. As a result, reducing dependence on ground-based facilities has become a key focus of research [5].

On 27 April 2021, the Qilu-1 satellite, which was developed by our colleagues, was launched with China’s first Ku-band SAR and Video-on-Demand (VOD) integrated payload, which is an X-band VOD communication payload for back-supportable application terminals [6]. Users in the satellite coverage areas would be able to control the SAR satellite in real time and to place mission instructions on the satellite. After the satellite completes remote sensing and graphics data in-orbit processing, it can transmit image data to users in real time, thus delivering an “ordering, watching, and transmitting” service. Given the commonality between the SAR satellite microwave payload and the VOD communication payload, the DFRC system can be used to reduce the number of single sets on the satellite, boost system integration and miniaturization, and improve spectrum utilization. The results verify the feasibility of the DFRC integrated system in space.

Dual-functional radar and communication (DFRC) technology, which combines radar and communication functions into one, has significant potential for a variety of applications [7]. This technology allows a single system to perform detection and SAR tasks while also executing functions like communication, data transmission, and navigation simultaneously. DFRC systems enable the transmission of radar and communication signals using one device at the same time with the same frequency band. This integration reduces the need for extra communication payload equipment and eliminates the requirement for additional communication frequency allocations, enhancing the efficiency of electromagnetic spectrum utilization [8,9].

The current DFRC research mainly focuses on studying joint modulation waveforms and radar application scenarios. Sahin et al. [10] suggested using continuous phase modulation and polyphase-coded frequency-modulated radar waveforms for transmitting information sequences. However, the efficiency of the communication signal is constrained by the radar’s constant envelope waveform, limiting the amount of information that can be transmitted. Chatzitheodoridi et al. [11] and Basit et al. [12] proposed a method using frequency division matrices for multi-input multi-output (MIMO) signals and adaptive MIMO signals. This approach utilizes a sidelobe modulation of communication signals to enable target tracking and communication functions within an integrated radar-communication system. However, the radar beams cannot cover all exploration areas simultaneously, leading to low energy efficiency. Yang et al. [13] proposed using a chirp-based multicarrier frequency division waveform for efficient radar detection and communication on the same frequency band. However, the simultaneous use of radar and communication signals causes co-channel interference, requiring additional research on interference suppression methods. Zhang et al. [14] used orthogonal frequency division multiplexing (OFDM) chirp waveforms to embed communication information in the chirp signals. They utilized band-pass filters to eliminate interference signals and successfully implemented MIMO-SAR with OFDM linear frequency modulation waveforms. However, the system was highly complex and required optimization for energy efficiency.

Meanwhile, research by Yang et al. [15] focuses on radar applications. They embedded a discrete Fourier transform-based watermarking framework into radar waveforms to enhance security and reduce the probability of signal interception. This enables stable communication and high-quality imaging in multipath and noisy environments. You et al. [16] designed a hybrid radar-communication system for estimating statistical channel state information. This system allows for data transmission in low Earth orbit satellite communication and target sensing abilities. Zhu et al. [17] and Liu et al. [18] successfully integrated radar and communication functions on spaceborne and ground-based SAR systems using FBMC. Compared to traditional OFDM waveforms, FBMC waveforms offer greater flexibility and higher spectral containment capability, making them a potential modulation waveform for DFRC [18]. Multipath effects arise from echoes originating from different reflecting objects. When the maximum delay of these echoes exceeds the length of the cyclic prefix (CP), inter-symbol interference occurs, disrupting the orthogonality between subcarriers. OQAM modulation shows excellent low sidelobe levels and resistance to multipath in integrated SAR and communication systems. By designing OQAM-OFDM transmission sequences, the need for a cyclic prefix, as in CP-OFDM, is eliminated, thereby enhancing spectral efficiency [19]. Channel estimation (CE) is one of the potential research objects of DFRC. For the radar-centered integration approach, Zhang et al. [20] proposed a tensor-decomposition method to estimate multipath channel parameters, including such complex parameters as azimuth and elevation. This method can reconstruct the wireless channel between any pair of transmitting and receiving mobile antennas and use the estimated multipath channel parameters to assist subsequent communication. Zhang et al. [21] proposed a method that uses the radar main lobe to estimate the high-moving channels of the perceived targets by utilizing prior knowledge on delays and Doppler frequency shifts at different frequencies. The communication performance of the communication-centric integration approach is generally not affected. When the receiver knows the transmitted waveform a priori, almost all of the communication waveforms can be used for target sensing. Li et al. [22] implemented channel estimation by using communication waveforms in massive MIMO systems, so as to perceive the information on target parameters and channel states. The framework is suitable for time-varying channels and frequency-selective channels and can be used to solve the squint effect of light beams in space. Li et al. [23] proposed adopting the multipath OFDM beams received by the passive radar system to estimate the target trajectory as well as the delay Doppler parameters and demodulation errors, so as to enhance the range and velocity resolution in the passive radar system, which would receive the communication and perceive the targets.

FBMC modulation signals have a high peak-to-average power ratio (PAPR) because of the extensive overlap of subcarriers. This overlap creates an unstable signal envelope, increasing the PAPR. In low linear amplification regions, this instability can cause significant nonlinear distortions, affecting both in-band and out-of-band signals. As a result, these distortions can degrade the bit error rate (BER) performance of FBMC waveforms and impact target detection capabilities [24]. Controlling the PAPR is essential when designing power allocation strategies.

Existing research on FBMC signals in SAR primarily focuses on investigating the ability of ground-based SAR systems to detect airborne targets [18,19]. In terrestrial applications, the communication system operates independently of the SAR system for message and data transmission. Therefore, integrated SAR and communication are less critical compared to spaceborne systems. Spaceborne SAR systems require communication capabilities to transmit imaging data back to Earth or to other satellites with less spectrum, power, equipment weight, and envelope than ground-based SAR systems. By integrating SAR and data transmission functions with DRFC technology, the satellite eliminates the need for additional resources like transmission spectrum and equipment space.

In order to achieve the FBMC-based integration of radar and data transmission, we propose a power allocation scheme for integrated spaceborne SAR and data transmission using FBMC-OQAM signals. We adjusted the transmission power of each subcarrier to maximize the SNR of SAR echoes while also ensuring the echo data transmission rate and adhering to PAPR constraints. The primary research focuses and innovations are as follows:

The integration of SAR satellite with data transmission systems has been achieved, allowing ground mobile terminals to obtain real-time SAR imagery data of their surrounding environment.

An algorithm has been designed for integrated spaceborne spotlight SAR and data transmission to achieve a balanced match between data transmission capacity and SAR imaging data volume. Given the limited power capacity of spaceborne SAR payloads, we aim to find an optimal power allocation strategy to meet power and PAPR constraints, while simultaneously optimizing performance and validating the simulation of SAR and data transmission systems.

In order to address the interference problem with SAR signals’ transmission and reception, we developed a Simultaneous Transmission and Reception (STAR) signal sequence based on spaceborne SAR echo and data transmission and reception. On the basis of the above functions, we designed an integrated system for SAR and data transmission, as well as a system workflow.

### 1.1. Design Concept

We propose a method that independently modulates radar and data transmission signals on different subcarriers within the same frequency band. By adjusting the power of each subcarrier separately, we can optimize the performance of both radar and data transmission systems simultaneously. Figure 1 illustrates an application scenario involving integrated spaceborne SAR and data transmission for single-target imaging. X, Y, and Z represent the range, azimuth, and elevation directions, respectively, with the system operating in a single-transmit–single-receive mode. The figure shows that the SAR satellite continuously transmits pulse signals and receives pulse echoes during the reception window. In stripmap and scan modes, the synthetic aperture time of SAR is relatively short, limiting the amount of data transmission and image resolution for mobile terminals. By using the spotlight mode, which continuously adjusts the antenna’s direction toward the target, the synthetic aperture time can be extended. This method enhances radar echo quality and supports large data transmission requirements.

During its flight, the SAR platform transmits a signal to the ground that combines SAR and data transmission functions. The platform can use the same antenna to receive echoes from ground targets for SAR imaging. It then sends the echo images as communication data to mobile terminals within the coverage area. This process uses spotlight mode to acquire echo information continuously and enable data transmission.

The SAR system’s signal, modulated by FBMC-OQAM, can be expressed as [25]:(1)$$x_{r}(t)=\sum_{k=1}^{K}\sqrt{p_{r,k}}q_{r}(t)e^{j2\pi (f_{c}+k\Delta f_{r})t+\phi (t)}=\sum_{k=1}^{K}\sqrt{p_{r,k}}\Phi_{r,k}(t)$$
where $x_{r}(t)$ represents the transmitted SAR signal; *K* denotes the number of subcarriers; $p_{r}={\left[p_{r,1},\ldots ,p_{r,K}\right]}^{T}$ indicates the power allocated to each subcarrier in the SAR system; $\Delta f_{r}$ is the subcarrier bandwidth spacing; ϕ(t) refers to the phase shift in the spotlight mode; $q_{r}$ refers to the prototype pulse modulation signal of FBMC-OQAM; and $\Phi_{r,k}(t)$ represents the modulated radar waveform. Using non-random sequence signals from data transmission systems for SAR imaging can compromise range ambiguity suppression in SAR imaging. Therefore, data transmission signals should not be used for SAR imaging in FBMC-OQAM systems.

The transmitted signal in an FBMC-OQAM data transmission system can be expressed as [26]:(2)$$x_{c}(t)=\sum_{k=1}^{K}\sqrt{p_{c,k}}d_{k}q_{c}(t)e^{j2\pi (f_{c}+k\Delta f_{c})t+\phi (t)}=\sum_{k=1}^{K}\sqrt{p_{c,k}}d_{k}\Phi_{c,k}(t)$$
where $x_{c}(t)$ represents the transmitted data signal; $p_{c}={\left[p_{c,1},\ldots ,p_{c,K}\right]}^{T}$ denotes the power allocated to each subcarrier in the data transmission system; and $d_{k}$ is the signal carrier symbol on the *k*-th subcarrier. To ensure consistent modulation, we use 4QAM modulation, which meets the condition $|d_{k}{|}^{2}=1$; $q_{c}$ refers to the prototype pulse modulation signal of FBMC-OQAM; $B_{w}$ indicates the signal bandwidth; $\Delta f_{c}$ represents the subcarrier bandwidth spacing; and $\Phi_{c,k}(t)$ represents the modulated waveform of the data transmission signal. The SAR signal reflects off the ground and is returned to the SAR satellite.

The echo signal, post matched filtering, can be expressed as:(3)$$y_{r,i}(t)=\beta_{i}x_{r}(t-\tau )e^{-j2\pi (f_{c}\tau )}+\omega_{r,i}$$
where $y_{r,i}(t)$ represents the demodulated baseband signal of the *i*-th SAR echo; *R* denotes the distance from the SAR satellite platform to the center of the target; *v* is the velocity of the satellite relative to the ground; τ signifies the time delay, with *c* being the speed of light and $\tau =\frac{2R}{c}$; $\beta_{i}$ represents the channel coefficient associated with the *i*-th subcarrier and the autocorrelation coefficient of the ground reflection coefficients; and $\omega_{r,i}$ is the Gaussian noise in the channel.

The data transmission signal received by the ground mobile terminal through OQAM demodulation is expressed as:(4)$$r_{c,i}(t)=\alpha_{i}x_{c}(t-\frac{\tau }{2})e^{-j2\pi (f_{c}\frac{\tau }{2})}+\omega_{c,i}$$
where $r_{c,i}(t)$ denotes the *i*-th demodulated baseband signal; $\alpha_{i}$ represents the channel coefficient associated with the *i*-th subcarrier and the autocorrelation coefficient of the ground reflection coefficients; and $\omega_{c,i}$ represents the Gaussian noise in the channel.

### 1.2. Signal Sequence Design

The concept of STAR (Simultaneous Transmission and Reception) is purposed to synchronously transmit and receive electromagnetic signals at the same space–time frequency; theoretically, it can improve the spectral efficiency and system capacity. STAR offers the possibility of simultaneous operation of SAR echo reception and data transmission systems; however, the transmitted signals may be leaked to the receiver through a low-noise amplifier, thus causing the problem of dramatic self-interference. The technologies to suppress self-interference can be divided into airspace self-interference suppression technologies, RF domain self-interference suppression technologies, and digital domain self-interference suppression technologies [27]. Jonathan et al. designed a digital phased array, in which the aperture is divided into a transmitter subarray and a receiver subarray; at the digital end, it can eliminate the interference of the received signals going into the low-noise power amplifier, thus achieving the isolation of the received and transmitted signals with a difference of 160 dB within a 100 MHz bandwidth [28]. In addition, Xu et al. [29] also eliminated interference from the transmitted signals by introducing filters in the low-noise power amplifier section. Figure 2 shows the integrated sequence of spaceborne SAR data transmission.

Based on the STAR method, this study designed a STAR integrated mode for the synchronous operation of SAR echo reception and data transmission; this mode is a kind of cooperation mode, where the data transmission signal and the SAR pulse transmission sequence are independent of each other. The integrated system divides the bandwidth into multiple subcarrier channels. The radar system uses a part of the subcarrier channel to transmit SAR signals, the transmission and reception of which are isolated from each other. The data transmission system uses the rest of the subcarrier channel to transmit data transmission signals separately. $T_{\mathrm{xr}}$ denotes the transmitting and receiving time of a single SAR image; $T_{\mathrm{PRI}}$ denotes the duration of SAR radar pulses; $T_{\mathrm{yr}}$ denotes the window time of receiving the echo; and $T_{\mathrm{xc}}$ denotes the data transmission time of data transmission signals. Among them, $T_{\mathrm{xc}}$ = $T_{\mathrm{xr}}$. For $T_{\mathrm{xc}}$, when the phased array receives the echo signals, the data transmission signals will be continuously transmitted. In addition, this mode has the following advantages: high data transmission rate, high spectral efficiency, and uninterrupted SAR imaging.

### 1.3. System Design

Figure 3 is an integration network for spaceborne SAR and data transmission based on FBMC. The satellite system includes a sequence controller, an OQAM modulator, an algorithm design, an FBMC modulator, imaging processing, a matched filtering of the SAR signals and data transmission spaceborne transmitter, and an echo SAR spaceborne receiver. The terminal system includes an FBMC demodulator and an image reconstruction from the data transmission receiver. The design scheme for the integrated waveform of the spaceborne SAR in spotlight mode and data transmission is as follows:

Step 1: The ground user submits a SAR imaging task request to the satellite via a mobile terminal.

Step 2: Upon receiving the user request, the sequence controller assigns a specific time slot for the SAR imaging task and configures the FBMC signal transmission plan based on the current task queue and resource availability.

Step 3: The satellite transmits SAR pulses, modulated by FBMC, toward the imaging target. The target reflects the SAR signals, generating echo signals that travel back through the echo channel.

Step 4: During the echo window period, the phased array ceases the transmission of SAR pulses, and the SAR satellite receives the echo signals using matched filtering.

Step 5: The SAR image data are extracted through image processing, followed by compression, encoding, and OQAM modulation, converting the information into data transmission signals intended for the ground terminal.

Step 6: The data transmission signal and the SAR signal for the next cycle undergo algorithmic power allocation and subcarrier channel assignment. They are then modulated using FBMC and transmitted through the wireless channel to the ground terminal.

Step 7: The mobile terminal receives the data transmission signals through its receiving antenna, demodulates them using FBMC, and reconstructs the image to obtain the required SAR image, thus completing the imaging task.

## 2. Construction of Mathematical Models

### 2.1. Data Transmission Weight

The size of a SAR echo image can be expressed as:(5)$$F=\frac{G\;A\;B}{\rho_{r}\rho_{a}}$$
where *F* represents the size of the echo image; *G* represents the compression ratio, which is determined by applying the high-compression JPEG-2000 method with a compressibility coefficient of *G* = 0.01; *A* indicates the imaging coverage area; *B* indicates bits per pixel of imaging figure; $\rho_{r}$ represents the range resolution; and $\rho_{a}$ signifies the azimuth resolution.

The data transmission capacity can be expressed as:(6)$$C_{s}=\Delta B\sum_{k=1}^{K}\log_{2}(1+SNR_{k})$$
where $C_{s}$ represents the transmission rate of the data signal and $SNR_{k}$ denotes the carrier-to-noise ratio of the *k*-th subcarrier signal. To simplify, the subcarrier bandwidth is given by $\Delta B=\Delta f_{c}=\Delta f_{r}=B_{w}/K$. The bandwidth of each subcarrier is higher in FBMC signals due to aliasing.

The data transmission weight of the integrated system can be expressed as:(7)$$\kappa =\frac{R_{b}T}{\Delta B}=\frac{F}{\Delta B}$$
where κ represents the average data transmitted by each subcarrier during the synthetic aperture time and *T* denotes the synthetic aperture time.

### 2.2. Modeling Based on Joint Allocation System

To optimize performance, each subcarrier is independently assigned to either the SAR system or the data transmission system.
(8)$$\begin{array}{l} C_{s}(p,u)=\Delta B\sum_{k=1}^{K}\log_{2}\left(1+\frac{|d_{k}{|}^{2}|\alpha_{k}{|}^{2}\|\Phi_{c,k}{\|}^{2}(1-u_{k})p_{k}}{\sigma_{c}^{2}}\right)\\ \;\;\;=\Delta B\sum_{k=1}^{K}\log_{2}\left(1+\frac{|\alpha_{k}{|}^{2}(1-u_{k})p_{k}}{\sigma_{c}^{2}}\right) \end{array}$$
where $C_{s}(p,u)$ represents the data transmission capacity and ***u*** is a binary selection vector [30], $u={\left[u_{1},\ldots ,u_{K}\right]}^{T}$, where $u_{k}\in \{0,1\}$. $u_{k}=0$ denotes that the data transmission signal uses the *k*-th subcarrier and $u_{k}=1$ indicates that the radar signal uses the *k*-th subcarrier. The power vector ***p*** represents the power of each subcarrier, given by $p={\left[p_{1},p_{2},\ldots ,p_{K}\right]}^{T}$. The radar power is given by $p_{r}=u⊙p$, while the data transmission power is $p_{c}=(1_{K\times 1}-u)⊙p$. $\sigma_{c}^{2}$ denotes the power of noise in the data transmission channel. Considering the normalization of the modulation waveform, we assume $\|\Phi_{k}{\|}^{2}=1$. SNR is a basic metric to measure the performance of radar echo, and it is also a common design quality in power allocation algorithms. This study does not involve the design of beamform or a precoding matrix, which correspond to different channel noise environments and reflect the direct relationship between subcarrier power and channels. On the other hand, PSLR and ISLR are related to the main lobe and sidelobe waveforms of the SAR signal itself, so they cannot directly reflect different channel noise environments. When the normalized backscattered signal intensity is set to 1, the SNR of SAR echo will be inversely proportional to the noise-equivalent sigma zero, which reflects the ability to detect the minimum backscattered signal [31]. The larger the SNR, the smaller the detectable backscatter signal becomes without being overwhelmed by noise, thereby enhancing recognition capability. Given that this study is focused mainly on the transmission quality of the overall SAR signals in channels, SNR is therefore used as the indicator for radar.
(9)$$SNR_{s}(p,u)=\sum_{k=1}^{K}\frac{|\beta_{i}{|}^{2}\|\Phi_{k}{\|}^{2}u_{k}p_{k}}{\sigma_{r}^{2}}=\sum_{k=1}^{K}\frac{|\beta_{i}{|}^{2}u_{k}p_{k}}{\sigma_{r}^{2}}$$
where $SNR_{s}(p,u)$ represents the SNR of SAR echoes and $\sigma_{r}^{2}$ denotes the power of noise in the imaging channel.
(10)$$Par_{s}(p,u)=\frac{\max(\|\Phi_{k}{\|}^{2}p_{k})}{\frac{1}{K}\sum_{k=1}^{K}\left(\|\Phi_{k}{\|}^{2}p_{k}\right)}=\frac{\max(p_{k})}{\frac{1}{K}\sum_{k=1}^{K}\left(p_{k}\right)}\leq \eta$$
where $Par_{s}(p,u)$ represents the PAPR of a channel and η indicates the constraint on the maximum PAPR. Given the fixed total power condition, the PAPR constraint can be reformulated as a peak constraint [32], and Equation (10) can be rewritten as follows:(11)$$\left\{\begin{array}{l} p_{k}\leq \frac{\eta P}{K}\\ \sum_{k=1}^{K}p_{k}\leq P \end{array}\right.$$
where *P* represents the total power of the transmitted signal.

The optimization problem $P_{1}$ can be formulated as:(12)$$\begin{array}{c} \left(P_{1})\underset{u,p}{\max}\;SNR_{s}(u,p\right)\\ \begin{array}{l} s.t.\left\{\begin{array}{l} \begin{array}{ll} \sum_{k=1}^{K}p_{k}\leq P & \;\;\;\;\;\;\;\;\;\;\;\;\;\;(12.a)\\ \xi \leq p_{k}\leq \frac{\eta P}{K} & \;(12.b) \end{array}\\ \begin{array}{ll} \sum_{k=1}^{K}\log_{2}\left(1+\frac{|\alpha_{k}{|}^{2}(1-u_{n})p_{k}}{\sigma_{c}^{2}}\right)\geq \kappa \;|u{|}_{1}=\kappa \sum_{k=1}^{K}u_{k} & \;(12.c)\;\\ u_{k}\in \{0,1\} & \;(12.d) \end{array} \end{array}\right. \end{array} \end{array}$$
where ξ represents the power allocation constraint, which is imposed to prevent all power from being allocated solely to the subcarriers with the highest SNR and ensure that subcarriers with the lowest SNR do not receive zero power [33], and $|u{|}_{1}$ indicates the number of subcarriers used for SAR. The lower bound of the data transmission capacity needs to match the data returned by $|u{|}_{1}$ radar subcarriers.

### 2.3. Algorithms

#### 2.3.1. Alternating Direction Method of Multipliers

The alternating direction method of multipliers (ADMM) is a highly efficient optimization technique, especially for decomposing large-scale problems. By utilizing alternating optimization and updating Lagrange multipliers, ADMM can attain the optimal solution. It is particularly effective for addressing problems with a separable structure, such as those that can be divided into two or more subproblems [34]. To solve the power allocation problem, we utilize the ADMM and penalty function constraint methods. By keeping other parameters constant, we convert the original non-convex problem into a convex one.

The binary constraint (12.d) is transformed into the following form:(13)$$u_{k}(1-u_{k})\leq 0$$

The optimization problem $P_{1}$ is transformed into:(14)$$\begin{array}{c} \left(P_{2})\underset{u,p}{\max}\;SNR_{s}(u,p\right)\\ \begin{array}{l} s.t.\left\{\begin{array}{l} \begin{array}{cc} \sum_{k=1}^{K}p_{k}\leq P & \;\;\;\;\;\;\;\;\;\;\;\;\;\;(14.a)\\ \xi \leq p_{k}\leq \frac{\eta P}{K} & \;(14.b) \end{array}\\ \begin{array}{ll} \sum_{k=1}^{K}\log_{2}\left(1+\frac{|\alpha_{k}{|}^{2}(1-u_{n})p_{k}}{\sigma_{c}^{2}}\right)\geq \kappa \;|u{|}_{1}=\kappa \sum_{k=1}^{K}u_{k} & \;(14.c)\;\\ u_{k}(1-u_{k})\leq 0 & \;(14.d) \end{array} \end{array}\right. \end{array} \end{array}$$

Constraint (14.c) is a non-convex constraint. By converting it into a penalty function for the objective function, the optimization problem $P_{2}$ is transformed into:(15)$$\begin{array}{c} \left(P_{3})\underset{u,p}{\max}\;SNR_{s}(u,p)+\rho G(u,p\right)\\ \begin{array}{l} s.t.\left\{\begin{array}{l} \begin{array}{cc} \sum_{k=1}^{K}p_{k}\leq P & \;\;\;\;\;\;\;\;\;\;\;\;\;\;(15.a)\\ \xi \leq p_{k}\leq \frac{\eta P}{K} & \;(15.b)\\ u_{k}(1-u_{k})\leq 0 & \;\;\;\;\;\;\;\;\;\;\;\;\;(15.c) \end{array} \end{array}\right. \end{array} \end{array}$$
where
(16)$$G(u,p)=\sum_{k=1}^{K}\left(\log_{2}\left(1+\frac{|\alpha_{k}{|}^{2}(1-u_{n})p_{k}}{\sigma_{c}^{2}}\right)-\kappa u_{k}\right)$$
where ρ serves as the penalty factor that ensures the convergence of the algorithm, setting ρ = 0.001, ensuring that the objective function adheres to the given constraint.

To simplify processing, the log2(·) function is replaced with the log(·) function. This modification does not affect the convergence results. In Equation (16), combining variables ***u*** and ***p*** within the log2(·) function complicates the solution process. To resolve this issue, an additional variable ***w*** is introduced. By using the method of Lagrange multipliers, the variables are successfully separated from the logarithmic function [35]. With fixed values of ***u*** and ***p***, the optimization problem $P_{3}$ can be transformed using the method of Lagrange multipliers as follows:(17)$$\begin{array}{c} (P_{4})\underset{w}{\max}\;\sum_{k=1}^{K}\rho \left(\log\left(1+w_{k}\right)\right)\\ s.t.\left\{\begin{array}{ll} w_{k}=\frac{|\alpha_{k}{|}^{2}(1-u_{n})p_{k}}{\sigma_{c}^{2}} & \;(17.a)\;\\ \frac{|\alpha_{k}{|}^{2}\xi }{\sigma_{c}^{2}}\leq w_{k}\leq \frac{\eta P|\alpha_{k}{|}^{2}(1-u_{n})p_{k}}{K\sigma_{c}^{2}} & \;(17.b) \end{array}\right. \end{array}$$

The Lagrange multipliers for Equation (17) are expressed as follows:(18)$$(L)(w,\nu )=\sum_{k=1}^{K}\rho \left(\ln\left(1+w_{k}\right)\right)-\nu_{k}w_{k}+\nu_{k}\frac{|\alpha_{k}{|}^{2}(1-u_{k})p_{k}}{\sigma_{c}^{2}}$$
(19)$$w_{k}=min(max(\frac{1}{\nu }-1,\frac{|\alpha_{k}{|}^{2}\xi }{\sigma_{c}^{2}}),\frac{\eta P|\alpha_{k}{|}^{2}(1-u_{k})p_{k}}{K\sigma_{c}^{2}})$$
(20)$$\nu_{k}=\nu_{k}+\mu (\frac{|\alpha_{k}{|}^{2}(1-u_{k})p_{k}}{\sigma_{c}^{2}}-w_{k})$$
where ν represents the Lagrange multipliers and μ denotes the penalty function used for the iterations in Equation (20). The problem $P_{4}$ is transformed into:(21)$$\begin{array}{c} \left(P_{5})\underset{u,w,p,\nu }{\max}\;\sum_{k=1}^{K}\frac{|\beta_{k}{|}^{2}u_{k}p_{k}}{\sigma_{r}^{2}}+G(u,w,p,\nu \right)\\ \begin{array}{l} s.t.\left\{\begin{array}{l} \begin{array}{l} \sum_{k=1}^{K}p_{k}\leq P\;\;\;\;\;\;\;\;\;\;\;\;\;\;\;\;\;\;(21.a)\\ \xi \leq p_{k}\leq \frac{\eta P}{K}\;\;\;\;\;\;\;\;\;\;\;\;\;\;\;\;\;(21.b) \end{array}\\ \begin{array}{l} w_{k}=min(max(\frac{1}{\nu }-1,\frac{|\alpha_{k}{|}^{2}\xi }{\sigma_{c}^{2}}),\frac{\eta P|\alpha_{k}{|}^{2}(1-u_{k})p_{k}}{K\sigma_{c}^{2}})\;\;\;(21.c)\;\\ \nu_{k}=\nu_{k}+\mu (\frac{|\alpha_{k}{|}^{2}(1-u_{k})p_{k}}{\sigma_{c}^{2}}-w_{k})\;\;\;\;\;\;\;\;(21.d) \end{array} \end{array}\right. \end{array} \end{array}$$
where
(22)$$G(u,w,p,\nu )=\rho \left(\ln\left(1+w_{k}\right)\right)-\nu_{k}w_{k}+\nu_{k}\frac{|\alpha_{k}{|}^{2}(1-u_{k})p_{k}}{\sigma_{c}^{2}}-\rho \kappa u_{k})$$

The steps for implementing ADMM with a penalty function are as follows (Algorithm 1):


**Algorithm 1** Alternating Direction Method of Multipliers**Input:** ξ, η, P κ**Output:** Channel allocation u; Power allocation p
 Initialization: Initialize $p^{\left(0\right)}$, $u^{\left(0\right)}$; α and β randomly, let t=0;  **repeat**  t=t+1;   With $p^{\left(t\right)}$, $u^{\left(t\right)}$ and $w^{\left(t\right)}$, update $u^{\left(t+1\right)}$ by solving (21);   With $p^{\left(t\right)}$, $u^{\left(t\right)}$ and $u^{\left(t+1\right)}$, update $w^{\left(t+1\right)}$ by solving (19);   With $u^{\left(t+1\right)}$, $u^{\left(t\right)}$ and $w^{\left(t+1\right)}$, update $p^{\left(t+1\right)}$ by solving (21);   With $u^{\left(t+1\right)}$, $p^{\left(t+1\right)}$ and $w^{\left(t+1\right)}$, update $\nu^{\left(t+1\right)}$ by solving (20); **until** Convergence


#### 2.3.2. Alternating Direction Genetic Algorithms

Genetic algorithms (GAs) simulate natural selection and genetics by iteratively searching a population of candidate solutions using crossover, mutation, and selection [36]. GAs are well-suited for solving complex nonlinear integer programming problems, especially in cases where the parameter space is vast and intricate [37,38].

This conversion also involves transforming non-convex constraints into convex ones. The iterative residual of the objective function determines when the iterations should terminate. The non-convex constraint (17) can also be relaxed using the convex relaxation method. $p^{t-1}$ represents the subcarrier allocation power in the previous iteration. Therefore, the Taylor series expansion of constraint (12.3) around $p^{t-1}$ is given by:(23)$$\sum_{k=1}^{K}\kappa u_{k}-\log_{2}\left(1+\frac{|\alpha_{k}{|}^{2}(1-u_{k})p_{k}}{\sigma_{c}^{2}}\right)\geq \sum_{k=1}^{K}\kappa u_{k}-\log_{2}\left(1+\frac{|\alpha_{k}{|}^{2}(1-u_{k})p_{k}^{t-1}}{\sigma_{c}^{2}}\right)\;+\frac{\left(1-u_{k})(p_{k}-p_{k}^{t-1}\right)}{\ln2\left((1-u_{k})p_{k}^{t-1}+\frac{\sigma_{c}^{2}}{|\alpha_{k}{|}^{2}}\right)}$$

Given a fixed power ***p***, $P_{2}$ is transformed into the following form:(24)$$\begin{array}{c} \left(P_{6})\underset{u}{\max}\;{\mathrm{SNR}}_{s1}(u\right)\\ \begin{array}{l} s.t.\left\{\begin{array}{l} \sum_{k=1}^{K}\kappa u_{k}-\log_{2}\left(1+\frac{|\alpha_{k}{|}^{2}(1-u_{k})p_{k}}{\sigma_{c}^{2}}\right)\;\geq \sum_{k=1}^{K}\kappa u_{k}\\ \;\;\;-\log_{2}\left(1+\frac{|\alpha_{k}{|}^{2}(1-u_{k})p_{k}^{t-1}}{\sigma_{c}^{2}}\right)+\frac{\left(1-u_{k})(p_{k}-p_{k}^{t-1}\right)}{\ln2\left((1-u_{k})p_{k}^{t-1}+\frac{\sigma_{c}^{2}}{|\alpha_{k}{|}^{2}}\right)}\;\\ u_{k}(1-u_{k})\leq 0 \end{array}\right. \end{array} \end{array}$$

Given the fixed values of u and $p^{t-1}$, $P_{2}$ can be expressed as:(25)$$\begin{array}{c} \left(P_{7})\underset{p}{\max}\;{\mathrm{SNR}}_{s1}(p|p^{t-1}\right)\\ \begin{array}{l} s.t.\left\{\begin{array}{l} \begin{array}{l} \sum_{k=1}^{K}p_{k}\leq P\\ \xi \leq p_{k}\leq \frac{\eta P}{K} \end{array}\\ \begin{array}{l} \sum_{k=1}^{K}\kappa u_{k}-\log_{2}\left(1+\frac{|\alpha_{k}{|}^{2}(1-u_{k})p_{k}}{\sigma_{c}^{2}}\right)\;\geq \sum_{k=1}^{K}\kappa u_{k}\\ \;\;\;-\log_{2}\left(1+\frac{|\alpha_{k}{|}^{2}(1-u_{k})p_{k}^{t-1}}{\sigma_{c}^{2}}\right)+\frac{\left(1-u_{k})(p_{k}-p_{k}^{t-1}\right)}{\ln2\left((1-u_{k})p_{k}^{t-1}+\frac{\sigma_{c}^{2}}{|\alpha_{k}{|}^{2}}\right)} \end{array} \end{array}\right. \end{array} \end{array}$$

As $P_{2}$ is a mixed-integer nonlinear optimization problem, a genetic algorithm is employed to alternately solve for the approximate solution of $P_{7}$ by fixing *u* and *p*. The alternating direction genetic algorithm is as follows (Algorithm 2):


**Algorithm 2** Alternating Direction Genetic Algorithm**Input:** ξ, η, P κ, Crossover probability $P_{cross}$, Mutation probability $P_{mutate}$ **Output:** Channel allocation u; Power allocation p
Initialization: Initialize population size: 50, max generations: 100, Tolerance: 1 × 10^−9^
$p^{\left(0\right)}$, $u^{\left(0\right)}$; α and β randomly, let t=0;   **repeat**   t=t+1;    According to fitness function (24) with $p^{\left(t\right)}$, update $u^{\left(t\right)}$, through crossover and mutation obtain $u^{\left(t+1\right)}$;    According to fitness function (25) with $u^{\left(t\right)}$, update $p^{\left(t\right)}$, through crossover and muttion obtain $p^{\left(t+1\right)}$;    Update p, u; **until**
t≥max generations


By solving $P_{5}$, $P_{6}$, and $P_{7}$, the integrated waveform allocation scheme ***u*** and the transmission power ***p*** for each subcarrier are obtained. In the next pulse, the optimized integrated waveform is transmitted. Using the most recent echo data, we gather information about the communication channel and recalibrate the adaptive integrated waveform. By adjusting the parameter κ, we balance the weights assigned to the radar and transmission systems to meet mission requirements.

## 3. Simulation Experiment and Result Analysis

We used the MATLAB simulation package in this experiment, with computational hardware consisting of an Intel Core i7-7700HQ CPU and a GTX 1060ti GPU. The effectiveness of the proposed method was validated using the simulation data package, with specific simulation parameters listed in Table 1.

The prototype filter used in the FBMC-OQAM system for the experiment is as follows:(26)$$q(t)=1+2\sum_{n=1}^{N-1}{\left(-1\right)}^{n}H_{n}\cos\left(\frac{2\pi nt}{T}\right)$$
where N represents the overlapping factor in FBMC and $H_{n}$ denotes an FBMC parameter, as detailed in Bellanger et al. [39].

Due to the high transmission power of spaceborne radar and communication systems, ground-based user receivers do not need high antenna gain. As a result, mobile terminals use omnidirectional antennas that are a quarter-wavelength of the signal carrier. The SAR satellite utilizes a planar phased array antenna, providing benefits such as convenient beamforming, flexible beam scanning, and reliable electronic control. These characteristics make it ideal for the wide scanning range necessary in SAR imaging [40]. The SNR for the downlink power of each subcarrier can be expressed as:(27)$${\mathrm{SNR}}_{k}=\frac{p_{k}h_{k}G_{\mathrm{sat}}G_{\mathrm{ut}}c^{2}}{{\left(4\pi f_{c}R\right)}^{2}N_{c}}={\alpha_{k}}^{2}p_{k}$$
where ${\mathrm{SNR}}_{k}$ represents the SNR for the downlink of the *k*-th subcarrier; $G_{\mathrm{sat}}$ and $G_{\mathrm{ut}}$ denote the satellite antenna gain and the receiving antenna gain, respectively; $N_{c}$ indicates the system noise power; and $N_{c}=k_{B}\Delta BT_{u}$. $h_{k}$ represents the stochastic channel parameter.

Based on Equations (5) and (7), the image data volume for SAR with 1 m azimuth resolution, 1 m range resolution, and a coverage area of 5 km by 15 km, using 14 and 15 subcarrier channels, ranged between 84 and 90 Mb. The synthetic aperture time was 1.97 s, requiring the data transmission system to achieve a transfer rate of 42.6 to 45.6 Mbps.

### 3.1. Simulation Result

In 100 Monte Carlo experiments, the BER of Rayleigh fading and additive white Gaussian noise (AWGN) channels at various SNR levels for CP-OFDM and FBMC modulation schemes are shown in Figure 4. The experiments were conducted under ideal conditions where Doppler effect and multipath effect were not considered.

The convergency performance of the objective functions of Algorithms 1 and 2 under different η at κ=0.5 is illustrated in Figure 5.

At η=6 dB, the convergency performance of the objective functions of Algorithms 1 and 2 under different κ is illustrated in Figure 6.

In the simulation, Algorithm 1 had an iteration time of 1.22 s, while Algorithm 2 took 6.52 s. Algorithm 1 achieved convergence in 2.44 s, whereas Algorithm 2 required between 39.12 and 52.16 s. For high-velocity low Earth orbit (LEO) SAR satellites traveling at approximately 7 km/s, Algorithm 1 is more suitable due to rapid changes in satellite-to-ground relationships and channel states. The short synthetic aperture time necessitates a fast algorithm speed to handle these dynamics, leading to the exclusive use of Algorithm 1 in subsequent experiments.

The equal power allocation scheme prioritizes communication by allocating superior channels to the communication system and assigning the remaining channels to the SAR system. Figure 7 illustrates the spectral efficiency versus total power for Algorithm 1’s allocation scheme and the equal power allocation scheme with a communication priority.

The two proposed improved algorithms can be compared with the classic power allocation algorithm. The water-filling algorithm [41] allocates power across different channels based on their specific states, following Shannon’s capacity formula. This algorithm prioritizes channels with lower noise power to maximize overall system performance. The greedy algorithm, as described in the literature [42], initially assigns data transmission power to meet required data rates. It then looks for the best power allocation for radar, considering the objective function parameters. Figure 8 illustrates the noise equivalent power of echo channels and the subcarrier power allocation for an integrated spaceborne SAR and data transmission system, both under a PAPR of η = 2 dB.

Figure 9 displays the performance of algorithms for the integrated spaceborne SAR and data transmission system. It shows the comparisons of echo SNR and frequency efficiency across various algorithms.

Table 2 displays the performance of quantitative analysis of different algorithms. We have defined as follows: Weighted score = Echo SNR + ρ * Spectral Efficiency.

### 3.2. Results Analysis

Figure 4 indicates that the BER of FBMC was slightly lower than that of OFDM at low SNR levels. When the power threshold ξ was set to 30 W, 40 W, and 50 W, the BER excluding channel encoding and decoding were $7.4\times 10^{-3}$, $2.4\times 10^{-3}$, and $7.4\times 10^{-4}$, respectively. Efficient channel encoding and decoding techniques, such as low-density parity-check codes or turbo codes, can be employed to significantly reduce the BER, often achieving a BER of $1\times 10^{-6}$ or lower. This ensures the reliability and integrity of data transmission, thereby meeting the BER requirements of satellite communication systems.

Figure 5 shows that Algorithm 1 reached its convergence value after the second iteration under varying maximum PAPR η. In Algorithm 2, a higher value of η leads to faster convergence. The larger the value of η = 6 dB, the faster the convergence. This suggests that allocating more power to subcarriers with better channel conditions expands the feasible solution space, making convergence easier. In Algorithm 1, the objective functions at 6 dB and 9 dB were similar, showing that the system’s objective function approached its upper limit at 6 dB under the constraint of ξ. As a result, changes in η had limited impact on objective functions. Algorithm 2 converged in eight iterations and may face scenarios where internal iterations exceed the maximum population iteration count, halting the process prematurely. Theoretically, higher values of η lead to faster convergence. However, at 6 dB, convergence may reach local optima, yielding slightly worse performance than Algorithm 2. Increasing population iterations can enhance convergence outcomes at the expense of longer computational time. In both algorithms, when η = 0 dB, power is distributed evenly among subcarriers, with the optimal power being P/K, resulting in suboptimal power allocation to better channels and lower objective function values compared to other η schemes. Normalized waveforms require increased PAPR in power allocation to boost the SNR of echo signals and algorithm convergence speed. The objective function results of Algorithm 1 outperform those of Algorithm 2 but require more iterations.

Figure 6 illustrates the convergence performance of the algorithm under varying transmission weights κ and its impact on SNR. The variable κ represents the weight of the task data volume for each transmission, as defined by Equation (7). When κ=0.5, it indicates the complete transmission of the entire data volume in each instance. When κ=0.25, it signifies the transmission of half the image data in each instance, while κ=0.05 corresponds to the transmission of one-tenth of the image data per instance. Algorithms 1 and 2 achieve convergence after two and six iterations, respectively. As the value of κ increases, the number of subcarriers required for communication also increases, which in turn leads to a degradation in radar performance. Consequently, the system needs to allocate more power and subcarrier channels to communication. In practical tasks, selecting κ=0.25 enhances the objective function and improves the overall radar and communication performance of the system compared to κ=0.5.

In Figure 7, at κ=0.5, $\kappa |u{|}_{1}$ represents the ratio of the data transmission volume required to the subcarrier bandwidth. The communication capacities of the two schemes of Algorithm 1 met the channel threshold and showed an improvement compared to the equal power allocation scheme. Compared to η = 3 dB, the communication capacity at η = 6 dB was closer to the channel threshold, resulting in greater power savings.

Based on Figure 8 and Figure 9 and Table 2, the genetic algorithm showed similar performance to the ADMM-based subcarrier allocation algorithm. The power allocation of the ADMM algorithm was the same as that of the greedy algorithm. The ADMM algorithm achieved the optimal SNR of echoes, while the genetic algorithm excelled in spectral efficiency but reduced the SNR of echoes. Although the ADMM algorithm did not greatly affect frequency efficiency, it improved the SNR of echoes. In general, the ADMM algorithm achieved the highest weighted score and displayed superior performance.

## 4. Conclusions

We proposed a unified design for power allocation methods in an integrated space-borne multicarrier radar and communication system. This method enabled independent subcarriers for communication and radar while sharing the same frequency band, RF transmission system, and transmitting antenna via FBMC-OQAM modulation. Algorithms 1 and 2, under certain PAPR constraints, successfully maximized the SNR of echoes while meeting data throughput requirements for transmission. This study explored the relationship between tolerable BER and required power in OFDM and FBMC communications. By utilizing two algorithms, we established a mathematical model that clarified the connections between peak power, image transmission weight, the SNR of echoes, and communication capacity. We then developed a strategy to enhance the SNR of echoes within specified limitations. Simulation results demonstrated that for images with 1 m azimuth and range resolutions, covering a 5 km by 15 km area, the required transmission rate for a single echo image ranged from 42.6 Mbps to 45.6 Mbps. In comparison to equal power allocation, Algorithm 1 enhanced the SNR of echo images by 2.79 dB and increased the data transmission rate by 24.075 Mbps under identical transmission.

## Figures and Tables

**Figure 1 sensors-24-06375-f001:**
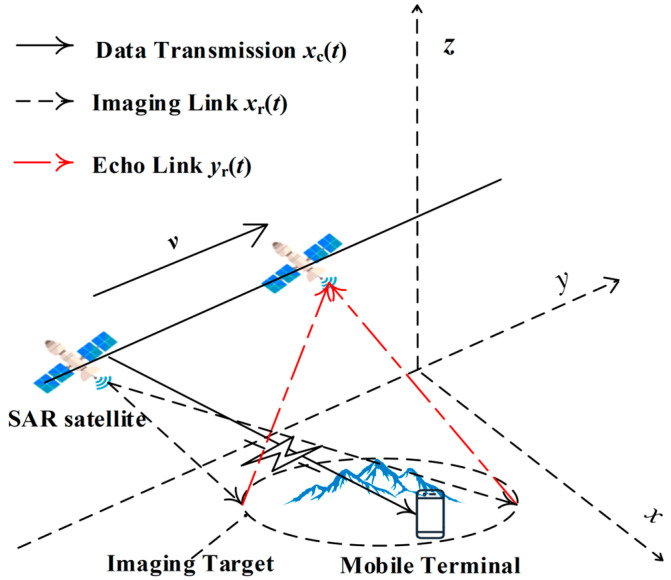
Schematic diagram showing the application scenario for the integrated spaceborne SAR and data transmission system.

**Figure 2 sensors-24-06375-f002:**
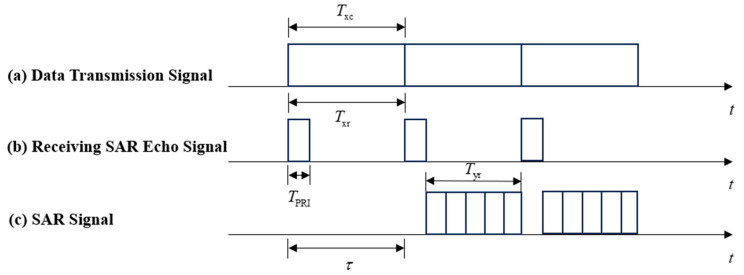
SAR echo reception and data transmission STAR sequence.

**Figure 3 sensors-24-06375-f003:**
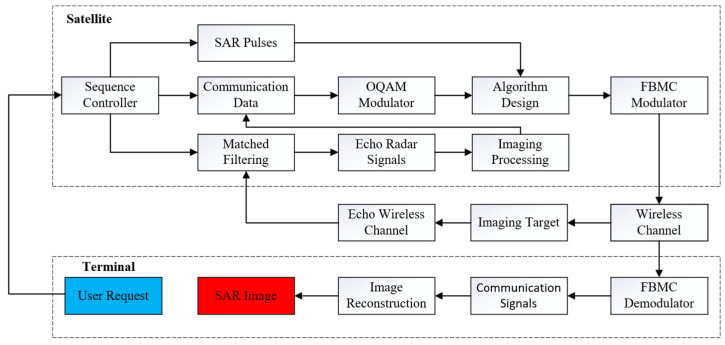
Integration network for spaceborne SAR and data transmission.

**Figure 4 sensors-24-06375-f004:**
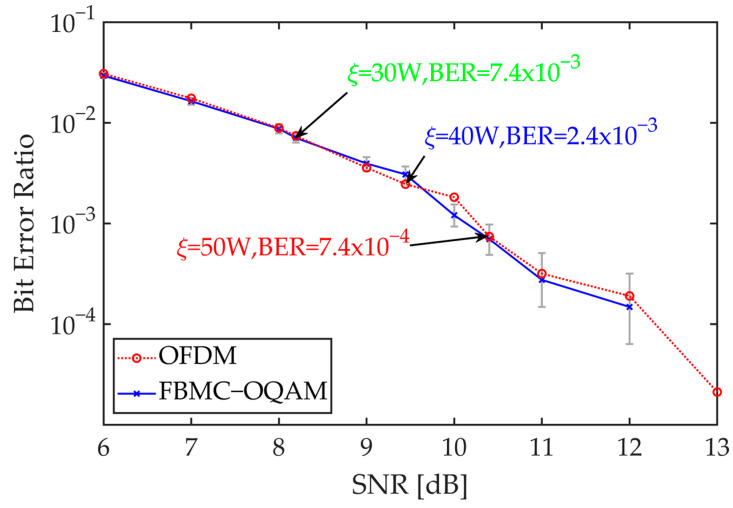
BER of Rayleigh fading and AWGN channels across various signal-to-noise ratio levels.

**Figure 5 sensors-24-06375-f005:**
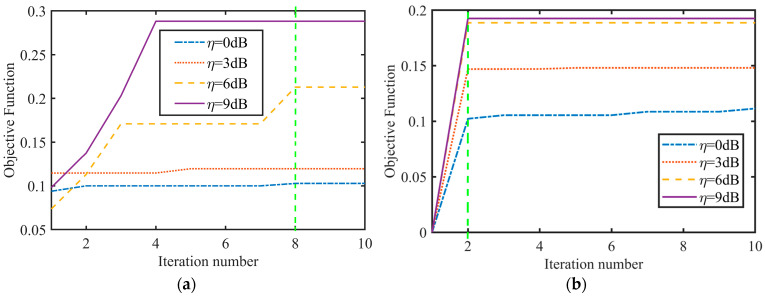
Convergence performance of the objective function under different values of η. (**a**) Algorithm 1. (**b**) Algorithm 2.

**Figure 6 sensors-24-06375-f006:**
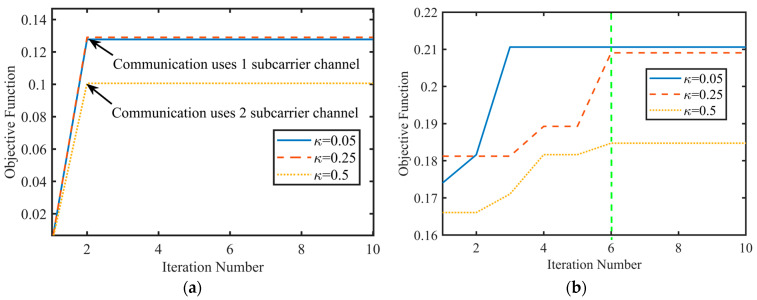
Convergence performance of the objective function under different values of κ. (**a**) Algorithm 1. (**b**) Algorithm 2.

**Figure 7 sensors-24-06375-f007:**
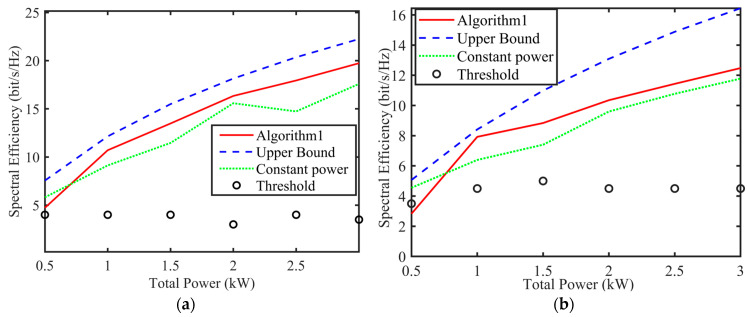
Spectral efficiency versus total power. (**a**) η = 6 dB scheme. (**b**) η = 3 dB scheme.

**Figure 8 sensors-24-06375-f008:**
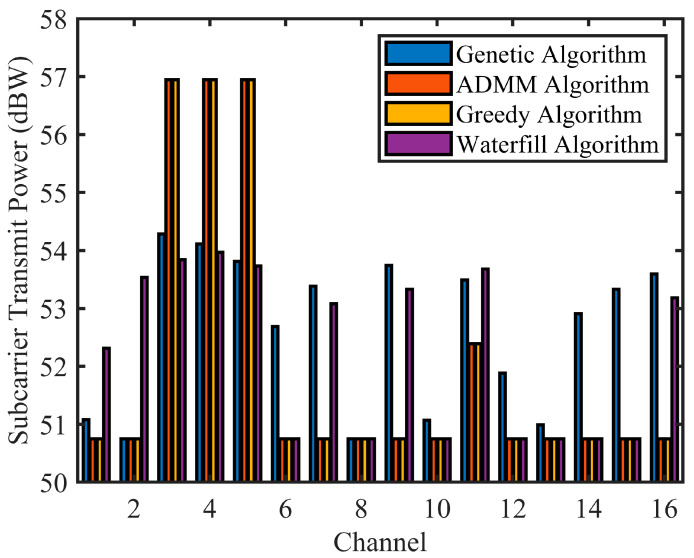
Subcarrier power allocation for integrated spaceborne SAR and data transmission system.

**Figure 9 sensors-24-06375-f009:**
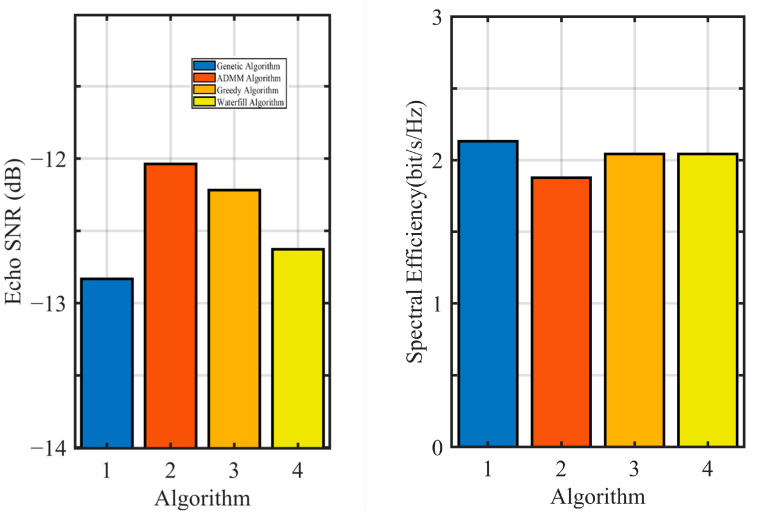
Algorithm performance for the integrated spaceborne SAR and data transmission system.

**Table 1 sensors-24-06375-t001:** Settings for simulation parameters.

	Parameters	Values
Satellite	$\mathrm{Bandwidth}\;B_{w}$	180 MHz
	$\mathrm{Carrier}\;\mathrm{frequency}\;f_{c}$	5 GHz
	$\mathrm{Sampling}\;\mathrm{frequency}\;f_{s}$	220 MHz
	Number of subcarriers *K*	16
	Orbital altitude *R*	500 km
	$\mathrm{Azimuth}\;\mathrm{resolution}\;\mathrm{of}\;a\;\sin\mathrm{gle}\;\mathrm{frame}\;\mathrm{image}\;\rho_{a}$	1 m
	$\mathrm{Range}\;\mathrm{resolution}\;\mathrm{of}\;a\;\sin\mathrm{gle}\;\mathrm{frame}\;\mathrm{image}\;\rho_{r}$	1 m
	Satellite velocity *v*	7 616 m/s
	$\mathrm{Squint}\;\mathrm{angle}\;\theta_{c}$	$0^{{}^{\circ}}$
	Synthetic aperture time *T*	1 s
Antenna	Antenna aperture *D*	1 m × 1 m
	RF peak power *P*	3000 W
	$\mathrm{Transmitting}\;\mathrm{antenna}\;\mathrm{gain}\;G_{\mathrm{sat}}$	35.43 dB
	$\mathrm{Mobile}\;\mathrm{terminal}\;\mathrm{antenna}\;\mathrm{gain}\;G_{\mathrm{ut}}$	1.76 dB
Other parameters	Power threshold ξ	30 W
	Coverage area *A*	5 km (azimuth) × 15 km (range)
	Peak-to-average power ratio η	0 dB, 1 dB, 3 dB, 9 dB
	Penalty parameter ρ	0.001
	Lagrangian penalty parameter μ	1 000
	Data transmission weight	0.05, 0.25, 0.5
	FBMC overlapping factor *N*	8
	$\mathrm{Boltzmann}\;\mathrm{constant}\;k_{B}$	$1.38\times 10^{-23}\;J\cdot K^{-1}$
	$\mathrm{System}\;\mathrm{temperature}\;T_{u}$	600 K
	$\mathrm{Crossover}\;\mathrm{probability}\;P_{\mathrm{cross}}$	0.5
	$\mathrm{Mutation}\;\mathrm{probability}\;P_{\mathrm{Mutate}}$	0.5

**Table 2 sensors-24-06375-t002:** Results for Algorithm performance.

Algorithm	Echo SNR	Spectral Efficiency	Weighted Score
Genetic Algorithms	−12.832 dB	4.262	0.0564
ADMM Algorithms	−12.037 dB	3.752	0.0663
greedy algorithm	−12.218 dB	4.085	0.0641
water-filling algorithm	−12.626 dB	4.085	0.0587

## Data Availability

Data are contained within the article.

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
