# Peer review of "Design of an Integrated System for Spaceborne SAR Imaging and Data Transmission"

_sensors, 2024, doi:10.3390/s24196375_

Round 1

Reviewer 1 Report

Comments and Suggestions for Authors

This manuscript tends to present a design of an integrated system for spaceborne SAR imaging  and data transmission. And there exists 8 major revisions Comments are listed as follows

Major revision 1: The SNR is not the criterion for SAR imaging performance. Authors should use IRW, PSLR, ISLR or image contrast, image entropy to measure SAR imaging simulation or real SAR data processing.

Major revision 2: Section 2.3 mentions that authors use the alternating direction method of multipliers, but in the table of Algorithm 1 presents alternating direction genetic algorithm. The writing logic of section 2.3 needs improvements.

Major revision 3: It seems that authors mainly solve conflicts between communication and SAR imaging with waveform design and optimum algorithm. And no system design is included. Maybe authors can not name the title with ‘system’.

Major revision 4: The summary of main innovation of this manuscript is not accurate. System design is not presented in this manuscript. Moreover, brief description on principle of the proposed method is not presented in the words of innovations. Authors should rewrite the innovation of this manuscript.

Major revision 5: It is hard for readers to find the conclusion of this manuscript from Fig6 to Fig.7. Authors should present more reader-friendly figures.

Major revision 6: Authors should present a flowchart of the proposed method for whole procedure.

Major revision 7: Authors should present Tables of quantitative analysis on experiments.

Major revision 8:  In experiment, which algorithm is the proposed method? And which algorithms are for comparison?

Comments on the Quality of English Language

 Minor editing of English language required.

Author Response

Thank you for the reviewer’s comments. All changes have been highlighted with gray shading.

Comments 1: [The SNR is not the criterion for SAR imaging performance. Authors should use IRW, PSLR, ISLR or image contrast, image entropy to measure SAR imaging simulation or real SAR data processing.]

Response 1: [Thank you for your suggestion. We have discussed which criterion would be more suitable for our algorithm in lines 304 to 314.]

Comments 2: [Section 2.3 mentions that authors use the alternating direction method of multipliers, but in the table of Algorithm 1 presents alternating direction genetic algorithm. The writing logic of section 2.3 needs improvements.]

Response 2: [We were really sorry for our careless mistakes. Thank you for your reminder. we have changed the site of “Algorithm1” with “Algorithm2” in line 375 and line 397.]

Comments 3: [It seems that authors mainly solve conflicts between communication and SAR imaging with waveform design and optimum algorithm. And no system design is included. Maybe authors can not name the title with ‘system’.]

Response 3: [We think this is an excellent suggestion and have added system part about signal sequence design and system design in line 213 to line 273.]

Comments 4: [The summary of main innovation of this manuscript is not accurate. System design is not presented in this manuscript. Moreover, brief description on principle of the proposed method is not presented in the words of innovations. Authors should rewrite the innovation of this manuscript.]

Response 4: [We sincerely apologize for the lack of clarity in our previous explanation, which led to a misunderstanding. We have revised the text for better clarity, and added clarifications between lines 14 to line 15 and line 149 to line153, as well as additional descriptions of the system]

Comments 5: [It is hard for readers to find the conclusion of this manuscript from Fig6 to Fig.7. Authors should present more reader-friendly figures.]

Response 5: [We think this is an excellent suggestion. We have explainde the change made, including the exact location where the change can be found in the revised manuscript between lines 528 and 534. We have added tables with quantitative analysis to enhance the readability of the paper in line 481.]

Comments 6: [Authors should present a flowchart of the proposed method for whole procedure.]

Response 6: [We have added a process description and a system structure diagram to help readers better understand the procedure in line 253 and line 272,.]

Comments 7: [Authors should present Tables of quantitative analysis on experiments.]

Response 7: [Thank you for your feedback. We have added tables with quantitative analysis to enhance the readability of the paper in line 481.]

Comments 8:  [In experiment, which algorithm is the proposed method? And which algorithms are for comparison?]

Response 8: [Algorithms 1 and 2 are the proposed improved algorithms, compared with Algorithms 3 and 4. We have added further descriptions in line 462.]

Reviewer 2 Report

Comments and Suggestions for Authors

In this work, the authors propose a method that aligns the data transmission rate with the imaging data volume. This approach balances SAR imaging performance with the requirements for real-time data transmission. To meet the need for mobile user terminals to access real-time SAR imagery data of their surroundings without depending on large traditional ground data transmission stations, the authors developed an application system based on filter bank multi-carrier offset quadrature amplitude modulation (FBMC-OQAM). This system enables SAR and data transmission signals to share the same frequency band, radio frequency transmission system, and antenna, creating an integrated sensing and communication system. The following issues are suggested to address.

1. In the work, the authors consider the dual function system using filter bank multi-carrier offset quadrature amplitude modulation. What is the performance advantage of this method over conventional OFDM signals?

2. How can the proposed method extend to the MIMO systems?

3. How the cyclic prefix affects the performance of the proposed method?

4. The authors may discuss the recently proposed methods in dual function radar and communications in the introduction, e.g.

[1] Integrated Sensing and Communication with Massive MIMO: A Unified Tensor Approach for Channel and Target Parameter Estimation

[2] Channel estimation for movable-antenna MIMO systems via tensor decomposition DOI:10.1109/LWC.2024.3450592

[3] Multi-target position and velocity estimation using OFDM communication signals

[4] Multidimensional Spectral Super-Resolution with Prior Knowledge with Application to High Mobility Channel Estimation

Author Response

Thank you for the reviewer’s comments. All changes have been highlighted with gray shading.

Comments 1: [In the work, the authors consider the dual function system using filter bank multi-carrier offset quadrature amplitude modulation. What is the performance advantage of this method over conventional OFDM signals?]

Response 1: [We think this is an excellent suggestion. On the one hand, FBMC waveforms require low orthogonality between subcarriers, which can counter Doppler and multipath effects. On the other hand, FBMC waveforms do not use Cyclic Prefix (CP), so the false targets can be avoided and spectral efficiency can be improved. We have explained the changes, including the exact location where the change can be found in the revised manuscript in line 91 to line 93. Our research is based on the foundation established in Reference [19].]

Comments 2: [How can the proposed method extend to the MIMO systems?]

Response 2: [That's a good suggestion. Both MIMO and phased arrays can deliver excellent performance for SAR systems. However, the aim of our paper is to reduce antenna equipment and miniaturize satellite payloads. Please refer to lines 42 to 52 for more details.]

Comments 3: [How the cyclic prefix affects the performance of the proposed method?]

Response 3: [We think this is an excellent suggestion. The cyclic prefix performs poorly in mitigating multipath effects and FBMC does not require a cyclic prefix (CP) and can still work successfully. We have explained the changes, including the exact location where the change can be found in the revised manuscript in line 92 to line 94.]

Comments 4: [The authors may discuss the recently proposed methods in dual function radar and communications in the introduction, e.g.

[1] Integrated Sensing and Communication with Massive MIMO: A Unified Tensor Approach for Channel and Target Parameter Estimation

[2] Channel estimation for movable-antenna MIMO systems via tensor decomposition DOI:10.1109/LWC.2024.3450592

[3] Multi-target position and velocity estimation using OFDM communication signals

[4] Multidimensional Spectral Super-Resolution with Prior Knowledge with Application to High Mobility Channel Estimation]

Response 4: [We sincerely appreciate the valuable comments. We have added background information on channel estimation (CE) and checked the literature carefully and added more references on [20] to [23] into the introduction part in the revised manuscript.]

Reviewer 3 Report

Comments and Suggestions for Authors

The manuscript proposes a method that adjusts the data transmission rate with the imaging data volume.

 An advantage of the manuscript is its comprehensive literature review. However, its structure has some issues, which are started in Section 2. Some of them are mentioned below:

1-     Equation (5), what is B?

2-     In Equation (5), Make space between G, A, and B or express them in a more proposer way to demonstrate the above parameters as three separate parameters in the definition of F.

3-     Line 199, sigma_c is represented as noise in data transmission. However, in Equation (8), we have (sigma_c)^2. Hence, in my view, (sigma_c)^2 should be defined, which is the power of noise.

4-     lines 202 and 204 repeat one expression. Delete one of them. This repetition is also found in lines 205 and 208. Again, delete one of them.

5-     Refer to expressions given in lines 214, 224,228, and 242. What are these?

6-     Equations 12, 14,15,16, and 17 should be defined better to represent good links with the correct expression in lines 214, 224,228, and 242.

7-     Lines 229, and 225, equations of (14) and (15) are the same. Update them.

8-     Line 232, how the parameter “rho” is defined? What is its values (range of its definition)?

9-     Update legends of Figures 3(a), 4(b) .

10- Put a legend for the left subplot of Figure 7. Define the “x-axis” of two graphs of this figure.

11- Put dot “.” At the end of the caption for each figure,

12- Sentences written in “conclusion should be in past time (rather than present).

Comments on the Quality of English Language

Please refer to the comments given in above.

Author Response

Thank you for the reviewer’s comments. All changes have been highlighted with gray shading.

Comments 1: [Equation (5), what is B?]

Response 1: [We feel sorry for our carelessness. In our resubmitted manuscript, the definition of B is supplemented in line 280. Thanks for your correction.]

Comments 2: [In Equation (5), Make space between G, A, and B or express them in a more proposer way to demonstrate the above parameters as three separate parameters in the definition of F.]

Response 2: [We think this is an excellent suggestion.]

Comments 3: [Line 199, sigma_c is represented as noise in data transmission. However, in Equation (8), we have (sigma_c)^2. Hence, in my view, (sigma_c)^2 should be defined, which is the power of noise.]

Response 3: [We sincerely thank the reviewer for careful reading. As suggested by the reviewer, we have corrected the“noise”into“the power of noise” in line 302 and line 316.]

Comments 4: [lines 202 and 204 repeat one expression. Delete one of them. This repetition is also found in lines 206 and 209. Again, delete one of them.]

Response 4: [We were really sorry for our careless mistakes. Thank you for your reminder.]

Comments 5: [Refer to expressions given in lines 214, 224,228, and 242. What are these?]

Response 5: [These is the process of improving the optimization problem.]

Comments 6: [Equations 12, 14,15,16, and 17 should be defined better to represent good links with the correct expression in lines 214, 224,228, and 242.]

Response 6: [We have re-written this part according to the Reviewer's suggestion. We have incorporated them into the algorithm in Section 2.3, rather than as part of the model.]

Comments 7: [Lines 229, and 225, equations of (14) and (15) are the same. Update them.]

Response 7: [We think this is an excellent suggestion. We have made the necessary changes in line 349.]

Comments 8: [Line 232, how the parameter “rho” is defined? What is its values (range of its definition)?]

Response 8: [We have added further descriptions in line 353. “rho” is constant in the algorithm design that ensures the convergence of the algorithm.]

Comments 9: [Update legends of Figures 3(a), 4(b).]

Response 9: [Thank you for your reminder. We have made the necessary changes in line 440 and line 445.]

Comments 10: [Put a legend for the left subplot of Figure 7. Define the “x-axis” of two graphs of this figure in line 477.]

Response 10: We have added a legend to the left subplot of Figure 7 and defined the x-axis for both graphs in this figure.

Comments 11: [Put dot “.” At the end of the caption for each figure,]

Response 11: [Based on your comments, we have made the corrections.]

Comments 12: [Sentences written in “conclusion should be in past time (rather than present).]

Response 12: [We have revised the sentences in the conclusion to use past tense, as requested.]

Round 2

Reviewer 1 Report

Comments and Suggestions for Authors

After revision, authors have improve this manuscript. But there still exists a serious conflict. Authors have wrong comprehension on SAR imaging performance. In this 2nd sentence of abstract, authors state that ‘This approach balances SAR imaging performance 10 with the requirements for real-time data transmission.’ SNR can measure the performance of radar echo, but it can not measure the performance of SAR imaging. With description on rows 302-314, it is obviously that authors is totally unfamiliar with SAR imaging. This conclusion ‘The SNR of SAR radar's echo will be proportional to the product of an image's range resolution and azimuth resolution[31].’ is totally wrong. If the  value of resolution is smaller, the resolution of SAR is better. If the value of SNR of echo is larger, the quality of signal is better. In this case, the SNR of SAR radar's echo can not be proportional to the product of an image's range resolution and azimuth resolution. Authors must solve this conflict on criterion of SAR imaging performance.

Comments on the Quality of English Language

Minor editing of English language required.

Author Response

We apologize for our misunderstanding. According to the formula (A-1) established in Ref [1] there seems to be a correspondence, we tried to establish a link between SNR and resolution to show that SNR is a metric that can be used to measure. Still, the resolution is also related to other factors in the formula, so our comprehension is wrong. Therefore, we use noise-equivalent sigma zero (NESZ) to show that SNR is feasible. NESZ is the ratio between the intensity of the backscattered signal from the target and the intensity of the system noise, reflecting the detection of the smallest backscattered signal. We set the normalized backscatter signal strength to 1. The NESZ is inversely proportional to the SNR, i.e., the larger the SNR is, the smaller the backscatter signal can be detected without drowning in noise, and the better the recognition capability. We have explained the changes made, including the exact location where the changes can be found, in the revised manuscript between lines 310 and 314.

[1] Doerry, A. W. Performance Limits for Synthetic Aperture Radar. Second Edition. 2005, 49.

Reviewer 2 Report

Comments and Suggestions for Authors

The reviewer recommends the acceptance

Author Response

We thank you for your thoughtful suggestions and insights. The manuscript has benefited from these insightful suggestions. I look forward to working with you and the reviewers to move this manuscript closer to publication in Sensors.

Reviewer 3 Report

Comments and Suggestions for Authors

All given contents were properly followed and fixed.

Author Response

(The authors gave the same response as above.)

Round 3

Reviewer 1 Report

Comments and Suggestions for Authors

It is obvious that authors are not familiar with SAR imaging. It is suggested that authors should weaken description on SAR imaging in this manuscript.

Comments on the Quality of English Language

Minor editing of English language required.

Author Response

Comment 1: It is obvious that authors are not familiar with SAR imaging. It is suggested that authors should weaken description on SAR imaging in this manuscript.

Response 1: We think this is an excellent suggestion and reduced the description on imaging performance throughout this manuscript. If there is anything else we should do, please do not hesitate to let us know.